# Structured in Space, Randomized in Time: Leveraging Dropout in RNNs for Efficient Training

**Anup Sarma**     **Sonali Singh**     **Huaipan Jiang**
**Rui Zhang**     **Mahmut T. Kandemir**     **Chita R. Das**
Pennsylvania State University
{avs6194, sms821, hzj5142, rmz5227, mtk2, cxd12}@psu.edu

## Abstract

Recurrent Neural Networks (RNNs), more specifically their Long Short-Term Memory (LSTM) variants, have been widely used as a deep learning tool for tackling sequence-based learning tasks in text and speech. Training of such LSTM applications is computationally intensive due to the recurrent nature of hidden state computation that repeats for each time step. While sparsity in Deep Neural Nets has been widely seen as an opportunity for reducing computation time in both training and inference phases, the usage of non-ReLU activation in LSTM RNNs renders the opportunities for such dynamic sparsity associated with neuron activation and gradient values to be limited or non-existent. In this work, we identify dropout induced sparsity for LSTMs as a suitable mode of computation reduction. Dropout is a widely used regularization mechanism, which randomly drops computed neuron values during each iteration of training. We propose to structure dropout patterns, by dropping out the same set of physical neurons within a batch, resulting in column (row) level hidden state sparsity, which are well amenable to computation reduction at run-time in general-purpose SIMD hardware as well as systolic arrays. We provide a detailed analysis of how the dropout-induced sparsity propagates through the different stages of network training and how it can be leveraged in each stage. More importantly, our proposed approach works as a direct replacement for existing dropout-based application settings. We conduct our experiments for three representative NLP tasks: language modelling on the PTB dataset, OpenNMT based machine translation using the IWSLT De-En and En-Vi datasets, and named entity recognition sequence labelling using the CoNLL-2003 shared task. We demonstrate that our proposed approach can be used to translate dropout-based computation reduction into reduced training time, with improvement ranging from $1.23\times$ to $1.64\times$, without sacrificing the target metric.

## 1 Introduction

Recurrent Neural Networks (RNNs) are an important class of machine learning approaches for analyzing sequential data including language, speech, time-series data, etc. Given an input training sequence, RNNs cells pass information from the previous time-step to the current time-step, in the form of a *hidden-state*, which helps the network to learn inherent temporal constructs. To further enhance the ability to learn from past information, Long Short-Term Memory(LSTM) [Hochreiter and Schmidhuber, 1997] cells have been proposed, which consist of multiple internal gates and a cell-state in addition to the hidden-state. Such LSTM cell-based RNNs have been successfully deployed in a variety of application domains, e.g., language modelling [Mikolov and Zweig, 2012], automatic speech recognition [Hannun et al., 2014, Amodei et al., 2016, Graves et al., 2013], sentiment analysis [Ma et al., 2018], and machine translation [Luong et al., 2015]. However, training such RNNs is a time- and compute-intensive task. This is because, for each input sequence, a

given RNN is logically unrolled into a sequence equal to the sequence length, and computation is performed following the dependencies. Once the training loss is calculated, it is back-propagated over the sequence in the reverse direction to compute the weight gradients, which is known as *back-propagation through time* (BPTT) [Werbos, 1990].

One of the promising ways of reducing the effective amount of computation is to exploit *sparsity* present in data structures. Sparsity refers to the fraction of zero-valued operands to the total size of the operands involved in a computation step. Thus, efficiently skipping zero-valued operands from computation leads to faster training time and energy efficiency. Convolutional Neural Networks (CNNs), owing to their prominent use of ReLU [Nair and Hinton, 2010] as the activation function, generate significant dynamic sparsity associated with the activation and neuron gradient values, which can be leveraged during training. However, due to the non-ReLU activation functions, e.g., *tanh* and *sigmoid*, dynamic sparsity in LSTM based networks is *not* inherent in nature. Such activation functions do not directly induce zero values, and hence the opportunity of exploiting sparsity is only limited to thresholding and approximation-based approaches [Gupta et al., 2019]. Although such techniques have been shown to work relatively well during the inference phase (*after* the network is fully trained), training still relies on high precision arithmetic operations. Also, the derivatives of the *tanh* and *sigmoid* functions do not lend themselves to sparsity as in ReLU. For the same reasons, the back-propagated gradient values are not sparse either, thus ruling out the opportunity for sparsity exploitation during the backward pass.

Motivated by these observations, in this work, we explore the opportunity of using the dropout-induced zero values as a source of sparsity in LSTMs. Dropout, a widely used regularizing technique, works by randomly dropping activation values in the forward pass. Since the dropped activation map becomes sparse in nature, we can take advantage of this sparsity towards achieving computation speed-up. However, such dropout masks are sampled randomly, which creates bottlenecks in accelerating the resultant sparse computation in a general-purpose SIMD hardware owing to the storage and memory access overheads. Therefore, we propose to structure such dropout patterns which makes it amenable for hardware acceleration. We distinguish between sparsity types, namely, *input* and *output* sparsity, depending on whether the input operands are zero-valued or an output operand is going to be zero. Since the dropout mask can be sampled ahead of time, exploiting output sparsity may seem straightforward. However, applying it naively to the hidden state can result in unintended consequences. This is because the sparsity also propagates to the cell states due to the inherent computational dependencies, resulting in possible loss of functional performance(e.g. accuracy). Further, we carefully analyze the computational flow in the backward pass of LSTM to identify sparse operands. We also extend the sparsity technique to SIMD GPU-based matrix-multiplication using shared memory, and show that that our proposed approach is exploitable in existing software based techniques. Our proposed sparsity pattern is also well-suited to be leveraged in systolic array-based computations, resulting in energy-efficiency benefits of dense matrix operations combined with high-performance sparse operations. In summary, this paper makes the following **key contributions**:

- We outline a generic framework for applying dropout in the context of LSTMs, considering uniformity of dropout patterns within a batch and across time steps. In addition to commonly used non-recurrent(NR) direction, we also extend dropout structural pattern to recurrent (NR+RH) directions to maximize the scope for potential training speedup.
- We provide detailed opportunities for exploiting the uniform sparsity in forward as well as backward passes of LSTMs based on sparsity propagation during training. We also identify the sparsity types based on input/output sparsity, and leverage it towards faster computation operation.
- We conduct extensive experiments with three representative applications including Language Modelling, Machine Translation, and Named Entity Recognition to evaluate the general applicability and performance benefits of our proposed approach. Our results show that uniform dropout pattern within a batch and random in time steps achieves significant training speedup ranging from 1.23x-1.64x, while not compromising on the target accuracy metric.

## 2 Related Work

**Dropout and DropConnect.** Several prior works study the scope of Dropout towards reducing model over-fitting and achieve regularization. Hinton et al. [2012] first proposed the idea of dropout, which randomly removes the neurons in a given layer of a feed-forward neural network for a given probability $p$. Wan et al. [2013] on the other hand proposed DropConnect, which, instead of neurons,

randomly drops the weight connections between two layers. Pham et al. [2014] and Zaremba et al. [2014] applied Dropout towards LSTM RNN regularization, where the dropout operator is specifically applied on the non-recurrent direction in LSTM. Kingma et al. [2015] and Gal and Ghahramani [2016] explored dropout strategies based on Bayesian Approximation. Gal and Ghahramani [2016] in particular also extended Dropout to the recurrent direction of LSTMs. Moon et al. [2015] have proposed using dropout on the LSTM cell-state instead of the hidden-states; however, their approach is application-specific towards automatic speech recognition (ASR) tasks and doesn't discuss generality. Semeniuta et al. [2016] investigated a variant of recurrent dropout that does not cause a loss of long term memory by letting the dropout mask operate directly on one of the internal state vectors. Krueger et al. [2016] proposed to regularize RNNs by randomly preserving hidden units from the previous time-step, instead of dropping them out. Merity et al. [2017] used DropConnect along hidden-to-hidden weight matrices as a form of recurrent regularization for LSTMs, while using Dropout on the non-recurrent dimension. However, all of these prior works focus on achieving better regularization to avoid over-fitting while our approach seeks to (a) act as a simple, plug-in replacement in existing LSTM applications without requiring to alter other hyper-parameter settings, and (b) reduce the overall training time by structuring the dropout pattern itself, while achieving the same degree of regularization effect.

**Weight Sparsity in Deep Neural Networks.** Sparsity in Deep Neural Nets has also been extensively explored, and the work in this area can be categorized as unstructured or structured types. In particular, unstructured pruning approaches [Han et al., 2015a,b, Dai et al., 2019, Mao et al., 2017, Narang et al., 2017a] result in random sparsity in weight matrices, which is difficult to accelerate on general-purpose hardware due to storage and memory access overheads. This has motivated structured sparsity-based approaches for both CNNs [Yu et al., 2017, Wen et al., 2016, Narang et al., 2017a] and RNNs [Lu et al., 2016, Wen et al., 2017, Narang et al., 2017b]. However, these works mainly aim to improve model size and speedup during model inference; in contrast, our work is aimed at achieving speedup during the training phase. We also believe our approach can be adapted in structured pruning approaches as well, towards faster training on general-purpose hardware.

## 3   A General Framework of Using Dropout for Efficient LSTM Training

In this section, we introduce a generic dropout framework to distinguish different possibilities of dropout application followed by the scope of the dropout-induced sparsity throughout different phases of training. We first highlight the computational dynamics of an LSTM cell [Hochreiter and Schmidhuber, 1997], which is a variant of RNN, consisting of four different types of gates:

$$\text{input gate}: \boldsymbol{i}_t = \text{sigmoid}(\boldsymbol{i}_t^*), \qquad \boldsymbol{i}_t^* = \boldsymbol{h}_t^{l-1}\boldsymbol{W}_i + \boldsymbol{h}_{t-1}^l\boldsymbol{U}_i + \boldsymbol{b}_i \qquad (1)$$

$$\text{forget gate}: \boldsymbol{f}_t = \text{sigmoid}(\boldsymbol{f}_t^*), \qquad \boldsymbol{f}_t^* = \boldsymbol{h}_t^{l-1}\boldsymbol{W}_f + \boldsymbol{h}_{t-1}^l\boldsymbol{U}_f + \boldsymbol{b}_f \qquad (2)$$

$$\text{output gate}: \boldsymbol{o}_t = \text{sigmoid}(\boldsymbol{o}_t^*), \qquad \boldsymbol{o}_t^* = \boldsymbol{h}_t^{l-1}\boldsymbol{W}_o + \boldsymbol{h}_{t-1}^l\boldsymbol{U}_o + \boldsymbol{b}_o \qquad (3)$$

$$\text{input modulation gate}: \boldsymbol{g}_t = \tanh(\boldsymbol{g}_t^*), \qquad \boldsymbol{g}_t^* = \boldsymbol{h}_t^{l-1}\boldsymbol{W}_c + \boldsymbol{h}_{t-1}^l\boldsymbol{U}_c + \boldsymbol{b}_c \qquad (4)$$

where $\boldsymbol{h}_t^l$ notation refers to hidden-state from layer $l$ and time step $t$. Based on the above four internal gate states, the following two output states are computed:

$$\text{cell state}: \boldsymbol{c}_t^l = \boldsymbol{f}_t \odot \boldsymbol{c}_{t-1} + \boldsymbol{i}_t \odot \boldsymbol{g}_t \qquad (5)$$

$$\text{hidden state}: \boldsymbol{h}_t^l = \boldsymbol{o}_t \odot \tanh(\boldsymbol{c}_t^l) \qquad (6)$$

Equations (1) through (4) form the core of an LSTM cell and are also the computationally intensive part of the operation, during both the forward and backward passes of training. Parameters $\{\boldsymbol{W}_*, \boldsymbol{U}_*, \boldsymbol{b}_*\}$ constitute the weight of the network, where $\boldsymbol{W}_*$ and $\boldsymbol{U}_*$ refer to the input-to-hidden and hidden-to-hidden weight projection matrices, respectively, and $\boldsymbol{b}_*$ is the bias. Each weight matrix is given by size $H \times H$, where $H$ is the LSTM hidden size.

### 3.1   Reframing the Scope of Dropout Structuring in LSTMs

Considering an LSTM cell with hidden state dimension $H$ and an input batch size of $B$, at each time step $t$, the cell generates a hidden-state matrix ($\boldsymbol{h}_t$) and cell-state matrix ($\boldsymbol{c}_t$), each of size $B \times H$.

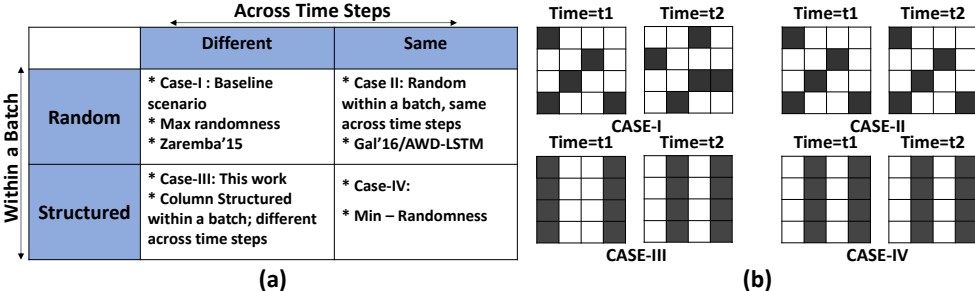

Figure 1: (a) A unifying sparsity framework based on dropout characteristics within a batch and across time steps. Based on two possible choices in each dimension, there are four possible cases. The prior works fall under case-I and case-II, while our proposed structured dropout mechanism belongs to Case-III. It allows taking advantage of structural sparsity due to dropout towards computation speed-up, while difference across time-steps leads to the necessary variation needed for achieving regularization objectives. (b) Graphical illustration of the four cases with dropout pattern applied to the two-dimensional hidden state matrix (dark squares indicate dropped out values).

Each row of this matrix corresponds to the feature vector of the corresponding input. The hidden state ($h_t$) is used as recurrent input to the next time-step $t+1$, and also as input to the subsequent LSTM cell/layer as a non-recurrent input.

The idea of regularizing recurrent neural networks by using dropouts has initially been introduced by Zaremba et al. [2014]. The two key aspects of the said work were – (a) dropouts were only used along the non-recurrent direction, and (b) at different time steps, it was sampled using different dropout masks making it time-variant. Subsequent work from Gal and Ghahramani [2016] has shown that dropout can be extended to the recurrent direction as well, thereby increasing network regularization scope. Similar to the work of Zaremba et al. [2014], the dropout mask itself was random. However, unlike Zaremba et al. [2014], the sampled dropout masks remained the same across time steps.

Given this possible choice of application of dropout, we propose a unifying framework considering the two dimensions of dropout pattern – dropout within a batch and dropout across time steps. Each dimension in turn offers two possible choices, resulting in four possible combinations, as summarized in Fig. 1(a) and illustrated in Fig. 1(b). Case-I refers to the scenario where dropout is random within a batch and also different across time-steps. This refers to the most common application of dropout including Zaremba et al. [2014]'s method. Case-II on the other hand refers to the scenario in which dropout pattern is random within a batch, but the same pattern is repeated across time steps. This approach is similar to Gal and Ghahramani [2016]'s work and has also been adopted later by Merity et al. [2017] in AWD-LSTM. Case-III refers to a dropout pattern that is structured within a batch, but varies across time steps. Finally, Case-IV is the most restricted form of dropout, where the pattern is not only structured within a batch, but also remains the same across time-steps.

These different dropout patterns exhibit different trade-offs. For example, while a mask, which is structured in a batch (B) and same across time-steps (T), is most amenable for hardware acceleration due to least metadata overhead, it also offers the least variability to the training model underneath. In this work, we stick to Case-III, where a structure in dropout mask within batch allows us to take advantage of computation speed-up, while variation across time-steps injects sufficient randomness into the training process, helping to retain the functional accuracy. Also, at a given time step, it is possible to view dropout being applied to only non-recurrent dimension (NR) or both non-recurrent as well as recurrent-hidden (NR+RH) dimensions. Therefore, we can further distinguish those cases as NR+ST or NR+RH+ST, and so on. In this study, we explore both NR+ST and NR+RH+ST configurations under Case-III towards maximizing potential training speedup, while retaining functional accuracy.

## 3.2 Dropout Propagation during Network Training

Next, we analyze the opportunity for dropout-based computation acceleration across the three key phases of network training: forward pass, backward pass, and weight gradient computation.

**Scope of Dropout Sparsity in Forward Pass (FP).**   During Forward Pass (FP), the most compute-intensive parts of LSTM cell are captured by Equations (1) through (4), in which inputs $\boldsymbol{h}_t^{l-1}$ and $\boldsymbol{h}_{t-1}^l$ are transformed by matrices $\boldsymbol{W}$ and $\boldsymbol{U}$, respectively. Since both recurrent and non-recurrent inputs to the cell would be structured-drop, this will result in the first input operand of the matrix multiplication being column sparse. Consequently, we can perform a sparsity-aware matrix-multiplication. This is shown in Fig 2(a). The resulting output matrix is a dense type on which subsequent reduction operations corresponding to Equations (5) and (6) could be performed.

Since dropout masks can be sampled ahead of time, ideally, we should also be able to apply the notion of *output sparsity* to prevent computation from taking place in the first place. Note however that, when output sparsity is applied to the hidden state, it is also inherently gets applied to the cell-state. This is because both the hidden state $h_t$ and cell state $\boldsymbol{c}_t$ is a combination of the same internal gate states. Therefore, any computation saving intended in the form of output sparsity for $\boldsymbol{h}_t$, also gets applied to the cell state $\boldsymbol{c}_t$. However, simply applying dropout to the LSTM cell state can be detrimental to the LSTM's ability to learn( Note that Moon et al. [2015] work to drop $\boldsymbol{c}_t$ doesn't warrant a generic deployment). Therefore, the scope of dropout sparsity exploitation in the forward-pass is only limited to *input type* in our studies, where the first input operand is column-sparse.

**Scope of Dropout Sparsity in Backward Pass (BP).**   To understand the feasibility of types of sparsity exploitation during the BP, we need to carefully take into account the effect of the back-propagation operator. In the case of FP, while input sparsity exploitation is straightforward, output sparsity is contingent upon applying dropout to the cell state as well. In the case of BP, the effect is closely related to the non-linearity used in Equations (1) through (4), which is $\mathrm{sigmoid}$ and $\tanh$ type. For both these non-linearity types, the derivatives are non-zero except at far ends from the origin, where derivatives tend to be zero asymptotically and are likely to result in random patterns of sparsity, if any. Therefore, we will look at the opportunity for sparsity exploitation due to the dropout operator from the FP.

We start by applying the back-propagation operator on the set of equations starting from Equation (6):

$$\delta\boldsymbol{o}_t = \delta\boldsymbol{h}_t \odot \tanh(\boldsymbol{c}_t), \qquad \delta\boldsymbol{c}_t = \delta\boldsymbol{h}_t \odot \boldsymbol{o}_t \odot (1 - \tanh^2(\boldsymbol{c}_t)) + \delta\boldsymbol{c}_{t+1} \tag{7}$$

Here notation $\delta\boldsymbol{x}$ refer to gradient of error with respect to variable $x$. Note that, in the above sets of equations, $\delta\boldsymbol{h}_t$ is structurally sparse, due to the application of the same dropout mask as in the forward pass on the set of computed gradient values. Thus, $\delta\boldsymbol{o}_t$ is sparse due to the only element-wise operator involving $\delta\boldsymbol{h}_t$. However, $\delta\boldsymbol{c}_t$ is not sparse in the same manner, in particular due to the recursive relationship involved.

The back-propagation operator for Equation (5):

$$\delta\boldsymbol{f}_t = \delta\boldsymbol{c}_t \odot \boldsymbol{c}_{t-1}, \qquad\qquad\qquad \delta\boldsymbol{c}_{t-1} = \delta\boldsymbol{c}_t \odot \boldsymbol{f}_t \tag{8}$$

$$\delta\boldsymbol{i}_t = \delta\boldsymbol{c}_t \odot \boldsymbol{g}_t, \qquad\qquad\qquad \delta\boldsymbol{g}_t = \delta\boldsymbol{c}_t \odot \boldsymbol{i}_t \tag{9}$$

Since $\delta\boldsymbol{c}_t$ is not sparse (in general), from the above expression, gradients arriving at different gate positions, i.e., $\delta\boldsymbol{f}_t$, $\delta\boldsymbol{i}_t$, and $\delta\boldsymbol{g}_t$ are also not sparse at this stage, except $\delta\boldsymbol{o}_t$.

Now, referring back to Equations (1) through (4), these gradients must back-propagate through the corresponding non-linearities resulting in $\delta\boldsymbol{i}_t^*$, $\delta\boldsymbol{f}_t^*$, $\delta\boldsymbol{o}_t^*$ and $\delta\boldsymbol{g}_t^*$. Therefore, the only sparse gradient operand $\delta\boldsymbol{o}_t$ also loses sparsity after back-propagating through the $\mathrm{sigmoid}$ activation. These non-sparse gradient values are used to compute the output gradient for $\delta\boldsymbol{h}_{t-1}^l$ and $\delta\boldsymbol{h}_t^{l-1}$ using the following methodology:

$$\delta\boldsymbol{h}_{t-1,i}^l = \boldsymbol{U}_i^\mathsf{T} \times \delta\boldsymbol{i}_t^*, \qquad \delta\boldsymbol{h}_{t,i}^{l-1} = \boldsymbol{W}_i^\mathsf{T} \times \delta\boldsymbol{i}_t^* \tag{10}$$

The above calculation step is also repeated with respect to the other three gates (Equations (2), (3), and (4)), and the corresponding hidden state gradients are added up in an element-wise fashion, before being passed through the same dropout mask used during the forward pass. Since the hidden state values eventually get dropped out before being passed to the previous state, we can take advantage of *output sparsity* here according to the dropout mask being applied. As shown in Figure 2(b), the output operator goes to zero according to the specific dropout pattern applied during the FP. Therefore, we can safely skip computing those locations by treating the second input operand column-sparse.

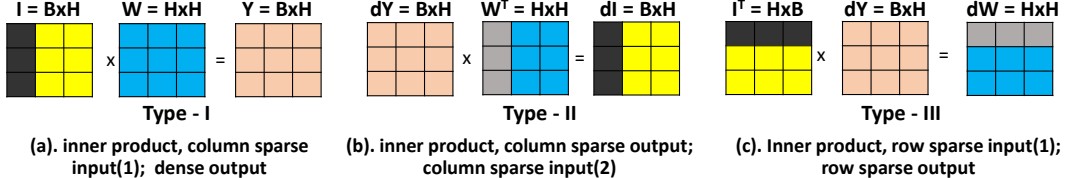

**Type - I**

**(a). inner product, column sparse input(1); dense output**

**Type - II**

**(b). inner product, column sparse output; column sparse input(2)**

**Type - III**

**(c). Inner product, row sparse input(1); row sparse output**

Figure 2: Resulting sparsity types due to dropout application during different stages of training.

**Scope of Dropout Sparsity in Weight Gradient (WG) Computation.** After computing the neuron gradients, it is now feasible to compute the weight gradients which are given by the inner product of the transposed input activation's map during the FP and input gradients during the BP. Thus, the weight gradient computation stage is applicable to the expressions involving $W$ and $U$ parameters from Equations (1), (2), (3), and (4). Corresponding to Equation (1), the weight gradient step can be written as follows:

$$\delta W_i = (h_t^{l-1})^{\intercal} \times \delta i_t^*, \qquad \delta U_i = (h_{t-1}^l)^{\intercal} \times \delta i_t^* \tag{11}$$

In both the expressions above, we observe that first input operands on the right hand side, i.e., $h_t^{l-1}$ and $h_{t-1}^l$, are column-sparse (from the FP application of structured dropout) operands. Therefore, we can leverage the notion of *input sparsity* with respect to these operands to accelerate computation in the WG computation phase. However, due to the application of the transposition operator, the first input operand for the matrix-multiplication becomes row-sparse as depicted in Figure 2(c). As a result, the generated output map also becomes row-sparse, correlating to the fact when a neuron is dropped during the FP, and it does not contribute to the WG computation in the BP.

In summary, the application of dropout creates opportunities for computation skipping throughout the LSTM training. Following the structured application of dropout pattern according to Case-III, the FP stage can support column-sparse input sparsity, the BP supports column-sparse output sparsity, and row-sparse input sparsity again during the WG computation stage. We next experimentally demonstrate the effectiveness of our proposed approach from the perspectives of both its regularization ability and training speed-up.

## 4 Experimental Results

Our proposed method can serve as a plug-in optimization in various applications using the LSTM-based models with dropout. We discuss the effectiveness and generality of our structured dropout approach by considering three representative application domains – Language Modelling (Section 4.1), Machine Translation (Section 4.2), and Sequence Labelling tasks (Section 4.3). Our training is conducted using an NVIDIA TITAN V GPU card hosted on a 12-core Intel Xeon Bronze 3104 CPU platform with Pytorch v1.8 as the training framework and CUDA version 11.2. The modified version of matrix multiplication operation is performed by skipping the loading of row and/or columns of respective input matrices into the GPU shared memory & register files according to dropout mask vector and type of sparse matrix operation as shown in Figure 2. In general, we refer to cases when only random dropout is applied in the non-recurrent direction as **NR+Random**, and the dropout pattern which is structured along non-recurrent direction as **NR+ST** (no recurrent dropout applied). Finally, **NR+RH+ST** refers to dropout which is applied to both the non-recurrent and recurrent directions and is also structured.

### 4.1 Language Modelling

**Model and Datasets.** We conduct language modelling experiments using the Penn Treebank (PTB) dataset [Marcus et al., 1993]. It consists of 929k training words, 73k validation words, and 82k test words, with a vocabulary size of 10k. We discuss the single model perplexity results corresponding from Zaremba et al. [2014] and AWD LSTM by Merity et al. [2017]. In each case, we keep the original training framework consistent, and the only modification we make is to change the pattern of the dropout application as discussed previously.

**Implementation Details.** Zaremba et al. [2014] experiment with two configurations: *medium* and *large*. Each configuration uses two layers of LSTM cells and a minibatch size of 20, and is unrolled

Table 1: Results of Language Modelling on Penn Treebank (PTB) showing perplexity on the validation and test sets, and the speedup for different training phases (FP, BP, and WG).

| Model | Perplexity | | Speedup | | | |
|---|---|---|---|---|---|---|
| | Validation | Test | FP | BP | WG | Overall |
| Zaremba et al. [2014] - Medium (Baseline) | 86.20 | 82.70 | 1 | 1 | 1 | 1 |
| This work(NR+ST) | 86.60 | 83.01 | 1.29 | 1.01 | 1.42 | 1.17 |
| This work(NR+RH+ST) | 83.64 | 80.84 | 1.66 | 1.10 | 1.57 | 1.45 |
| Zaremba et al. [2014] - Large (Baseline) | 82.20 | 78.40 | 1 | 1 | 1 | 1 |
| Krueger et al. [2016]-Large | 81.40 | 77.40 | 1 | 1 | 1 | 1 |
| This work (NR+ST) | 85.42 | 82.39 | 1.47 | 1.09 | 1.31 | 1.27 |
| This work (NR+RH+ST) | 79.38 | 76.05 | 2.45 | 1.28 | 1.41 | 1.64 |
| Merity et al. [2017] AWD-LSTM (Baseline) | 63.31 | 61.21 | 1 | 1 | 1 | 1 |
| This work(NR+RH+ST) | 63.95 | 61.77 | 1.63 | 1.04 | 1.53 | 1.38 |

for 35 time steps. For *medium*, each LSTM layer consists of 650 hidden units with parameters uniformly initialized in [-0.05, 0.05]. A 50% dropout is applied only on the non-recurrent dimension of LSTM cells, which is essentially random in nature and also varies across time-steps. After 39 epochs of training, this baseline network achieves validation and test perplexity results of 86.2 and 82.7, respectively. For *large*, each LSTM layer has 1500 hidden units uniformly initialized in [-0.04,0.04]. The *large* LSTM variant also employs a larger dropout pattern of 0.65 (random) along the non-recurrent direction. The large regularized LSTM achieves validation and test perplexity values of 82.2 and 78.4, respectively, after 55 epochs of training. Our dropout modifications to these networks involve (a) structuring the existing dropout levels (50%/65%) on the non-recurrent direction, and (b) introducing an additional (50%/65%) structured dropout along the recurrent direction. For the large model, we also report the corresponding results from Krueger et al. [2016] which improves Zaremba et al. [2014] test perplexity from 78.4 to 77.4.

We also experiment with Merity et al. [2017]'s ASGD Weight Dropped LSTM(AWD-LSTM), which consists of a three-layer LSTM with an embedding size of 400, and each LSTM hidden layer has a size of 1150. The LSTM weight parameters are initialized uniformly between $[-\frac{1}{\sqrt{H}}, \frac{1}{\sqrt{H}}]$, where $H$ is the LSTM cell size, while all embedding weights are initialized in [-0.1, 0.1]. For AWD-LSTM, the dropout rate applied at different levels of networks varies, which can be expressed by dropout vector of [0.4, 0.1, 0.25, 0.4] corresponding to input dropout, embedding layer dropout, non-recurrent LSTM dropout, and output dropout(pre-FC layer). In addition, it applies a dropout value of 0.5 on the recurrent weight matrices. Note that when we structure the recurrent weight matrix dropout, it essentially becomes structured dropout applied on the recurrent feature maps due to the computational symmetry of matrix multiplication.

We also analyze Gal and Ghahramani [2016]'s proposed approach in regularizing recurrent direction on top of Zaremba et al. [2014]'s prior work. However, due to the lack of a consistent framework to replicate these results (original Lua implementation), we avoid further quantitative comparison. Also, Moon et al. [2015] PTB language modelling network using 256 hidden units per layer is not compatible with either Zaremba-medium or large configurations, and also has higher perplexity numbers as compared to Zaremba-medium baseline. So, we omit these comparisons as well.

**Results and Discussion.** Table 1 shows that our NR+RH+ST modifications improve the validation and test perplexity levels to 83.64 and 80.84, respectively, for the medium configuration. Similarly, we observe an improvement in the case of the large LSTM configuration as well, with validation and test perplexity scores reaching 79.38 and 76.05, respectively, which also represents an improvement over Krueger et al. [2016]. For AWD-LSTM, our proposed optimization yields comparable validation and test perplexity with respect to the baseline after 100 epochs of training.

Table 1 also shows the speedups achieved on these particular set of benchmarks, with a breakdown of speedup during the FP, BP and WG stages of training. Our proposed structured dropout mechanism in the recurrent direction results in an overall speedup of 1.45× for the Zaremba-medium configuration, and 1.64× for the Zaremba-large configuration, while providing improved/competitive perplexity levels with respect to the state-of-the-art baseline. For the AWD-LSTM workload, the observed speedup

Table 2: Results of Machine Translation on IWSLT'14 De-En and IWSLT'15 En-Vi.

| Model | BLEU | | Speedup(De-En/En-Vi) | | | |
| --- | --- | --- | --- | --- | --- | --- |
| | De-En | En-Vi | FP | BP | WG | Overall |
| Luong et al. [2015] | 28.13 | 25.26 | 1 | 1 | 1 | 1 |
| This work (NR + ST) | 27.54 | 26.08 | 1.17/1.16 | 1.13/1.01 | 1.22/1.14 | 1.17/1.09 |
| This work (NR + RH + ST) | 28.46 | 26.20 | 1.35/1.33 | 1.17/1.07 | 1.45/1.37 | 1.31/1.23 |

is 1.38× over the baseline. Intuitively, this implies that we can achieve significant speedups (in the range of 45% to 64%) in the training of these workloads, while maintaining almost similar accuracy. For the Zaremba-medium and large cases, we also report the perplexity and speedup metrics obtained by running NR+ST configuration. As shown in the table, for both the configurations, the test and validation perplexity values are marginally worse than their baseline, while speedup-improvements – 1.17x for medium and 1.27x for large – are also lower than NR+RH+ST case, which is as expected.

Another important point that can be observed from this table is that each of the phases results in a different amount of speedup, although the total computation size and sparsity fraction are the same. More specifically, we note that the FP and WG phases have higher speedups compared to the BP phase. This essentially stems from the fact that sparsity is applicable for different matrix multiplication operands (refer to Fig. 2) in different phases, resulting in differently-shaped MM operations, which in turn leads to the difference in speedup.

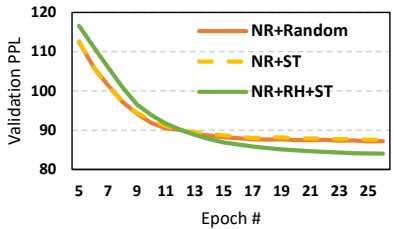

Figure 3: Perplexity during training.

Fig. 3 shows an interesting trend with respect to the validation perplexity during the course of training, corresponding to the Zaremba-Medium configuration. The plot shows that our proposed NR+RH+ST scheme starts with a higher level of perplexity as compared to Baseline(NR+Random) and NR+ST configurations. However, as the training proceeds, the perplexity continues to get lower for the NR+RH+ST case, while the Baseline and NR+ST cases start to get flatten. This further indicates the effectiveness of regularization achieved in our final approach.

## 4.2 Machine Translation

**Model and Datasets.** We evaluate our framework on two machine translation datasets, IWSLT'14 German-English (De-En) and IWSLT'15 English-Vietnamese (En-Vi) parallel corpus. After pre-processing and data-cleanup, IWSLT'14 De-En results in 153k sentence pairs, while IWSLT'15 En-Vi results in 133k training sentence pairs. For development and testing, we utilize the 2012 and 2013 test sets for the language pairs, respectively. We experiment with the NMT model proposed by Luong et al. [2015] that consists of a 2-layer unidirectional LSTM encoder-decoder with attention.

**Implementation Details.** Each LSTM layer consists of 512 hidden units and a maximum vocabulary size of 50,000 for both the source and target languages. It uses a batch size of 64, and a dropout probability of 0.3 (only in the non-recurrent direction) for both the encoder and decoder. The evaluation metric is bi-lingual evaluation understudy (BLEU), which is evaluated on the development set at each 5K training steps. Our implementation is based on the popular OpenNMTPy [Klein et al., 2017], a Pytorch-based framework for training NMT models. In this set of experiments, we run each dataset for a total of 65k steps under the baseline configurations. Without affecting any other hyper-parameter setting, we first structure the existing dropout pattern along the non-recurrent direction. Then, we insert an additional structured dropout of 0.3 along the recurrent direction as a part of enhancing the computation reduction opportunities. To further enhance it, we added dropout layers (0.3) to the final output of the encoder and decoder layers.

**Results and Discussion.** Table 2 shows the results corresponding to the baseline and our proposed approach. The baseline model achieves a BLEU score of 28.13 and 25.26 for the De-En and En-Vi datasets, respectively. Our NR+ST extension achieves BLEU scores of 27.54 and 26.08, while achieving speedups of 1.17x and 1.09x, for the two datasets. Our further modifications involving

Table 3: Results for Named Entity Recognition on the CoNLL-2003 shared task.

| Model | NER Metrics | | | | Speedup | | | |
|---|---|---|---|---|---|---|---|---|
| | Acc. | Prec. | Recall | F1-score | FP | BP | WG | Overall |
| Ma and Hovy [2016] | 97.57 | 89.29 | 89.23 | 89.26 | 1 | 1 | 1 | 1 |
| This work (NR + ST) | 97.67 | 89.86 | 89.46 | 89.32 | 1.43 | 1.06 | 1.18 | 1.21 |
| This work (NR + RH + ST) | 97.64 | 89.34 | 89.64 | 89.49 | 1.70 | 1.20 | 1.32 | 1.39 |

NR+RH+ST yield BLEU scores of 28.46 and 26.20 for the same two datasets, indicating that our proposed dropout extension is not only able to preserve the original learning dynamics, but also marginally improve upon it. Table 2 also shows the corresponding speedups achieved for the two datasets, which are 1.31x for De-En and 1.23x for En-Vi. Although the model and dropout configurations are the same for both the datasets, the difference arises in the decoder's final projection FC layer configuration, which is unique for each language depending on its vocabulary size.

### 4.3 Sequence Labelling for Named Entity Recognition

**Model and Datasets.** In this part, we experiment with a sequence labelling task for Named Entity Recognition (NER) to locate and classify named entities in an unstructured text. We refer to the work from Ma and Hovy [2016], which uses Bidirectional LSTM-CNNs-CRF towards achieving state-of-the-art results in sequence labelling. The proposed model architecture consists of a convolutional neural network (CNN) layer to encode the character-level information of a word into its character-level representation. Next, the character and word level representations are combined and fed as an input into the bi-directional LSTM layer. Finally, a sequential CRF is used to jointly decode the labels for the entire sentence. We perform our experiments using the CoNNL 2003 shared task dataset, which consists of four different types of named entities. The total number of word tokens for training, development and test in this dataset are 204k, 51k and 46k, respectively.

**Implementation Details.** We use the publicly available framework [Chernodub et al., 2019] to refer to the baseline and implement our modifications on top of it. This is a single-layer bidirectional LSTM network, and dropout was only applied at the input level. More specifically, input to the CNN layer is applied with a dropout of 50% (random) and its output is concatenated with embedding layer output, which is dropped at 50%. Since the CNN output contributes more towards the overall input feature dimension, the input to the BiLSTM layer is only $\sim$12% sparse. In our setting, we move the dropout layer from the input of the CNN layer to the concatenated output, thereby increasing the input sparsity level to 50%. In addition, we apply 50% structured dropout to the recurrent dimension of both the forward and reverse directions of BiLSTM, which is originally not present. Training results are shown for a maximum of 60 epochs.

**Results and Discussion.** Table 3 shows the values of the different evaluation metrics associated with this application for the baseline and the modified scenario. It can be observed from these results that, our proposed modifications with both NR+ST and NR+RH+ST fare equal or marginally better than the baseline configuration across the accuracy, precision, recall, and F1-score metrics. The proposed scheme also leads to an overall speed-up of 1.21x and 1.39x over the baseline corresponding to NR+ST and NR+RH+ST schemes, which further highlights the efficacy of our approach.

## 5 Conclusion

In this work, we holistically review and analyze the scope of leveraging dropouts induced sparsity for computation reduction of LSTM-RNN based neural network training. We explore the potential of structuring both the non-recurrent and recurrent dimensions associated with neural networks, which can bring potential speed-up during all phases of network training involving forward, backward, and weight-gradient computation stages. We also systematically outline the scope and nature of sparsity patterns in each phase. We validate the efficacy of our proposed approach by applying it to three popular and broad classes of NLP tasks targeting state-of-the-art GPU stack, and show that it achieves nearly identical(or even better in cases) accuracy compared to the baseline scenario, with significant execution speedup, ranging from 1.23x to 1.64x, in all three NLP tasks.

## ACKNOWLEDGEMENT

This research was supported in part by NSF grants #1317560, #1955815 and #1763681.

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
