# OpenReview forum: "Structured in Space, Randomized in Time: Leveraging Dropout in RNNs for Efficient Training"
_NeurIPS.cc/2021/Conference — NeurIPS 2021 Poster_

### Official Review · Reviewer_gJ9w · 2021-07-14

**Rating:** 6
**Confidence:** 4

**Summary:**

The authors present a generic framework to applying dropout as a sparsity technique to LSTM to reduce training time and as a plug-in technique w/o hyperparameters adjustment when comparing to previous similar techniques that focus on regularization. The dropout technique is randomized in timesteps but structured in the batch as a dropout mask that saves computation time on the dropout mask (does not compute), as well as inducing the regularization effect. This technique is applied to both the non-recurrent part of LSTM and the recurrent-hidden layer.

**Main Review:**

The paper is well written with a nice literature review. The author also provides a clear discussion about the difference of the proposed method and other literature. The computation acceleration across forward, backward, and gradient computation is well analyzed and verified with meaningful experiments.

Testing on different types of GPUs could make the results more convincing.

Minor things:
Line 72, remove extra "that"
Line 164, "ST" appears for the first time, please define it.


**Time Spent Reviewing:**

3

---

> ### Author Response · Authors · 2021-08-10
> **Testing on different GPU types**
>
> Thanks for the suggestion. We were partially able to conduct experiments(Language Modelling) on an alternate GPU platform (Tesla K20m with CUDA 7.5) and observe speedup improvement.
>
> For Tesla k20m,  which has lower clock frequency and throughput, we observe that speedup improvements from dense to sparse execution becomes even more prominent. This difference in speedup from dense to sparse can be up to ~20% higher in K20m, as compared to Titan V based execution. This indicates that our proposed solution is effective in existing as well as legacy GPU hardware generations.
>
> We will add specific and completed results in our final version under the appendix section.

---

### Official Review · Reviewer_nstS · 2021-07-15

**Rating:** 6
**Confidence:** 4

**Summary:**

The paper proposes a method to speed up training the LSTM networks. By introducing the structure into the dropout operation, authors are able to use sparse matrix multiplications. The paper carefully classifies different approaches drop out recurrent and feed-forward connections in LSTMs. Based on this classification, the paper chooses an approach to drop out both feed-forward and recurrent connections to get the most benefit from the sparse operations.

Second, the paper conducts experiments on series of tasks showing similar performance as baselines with speed-ups in training.

**Limitations And Societal Impact:**

The work is missing the discussion on the limitations of the approach. Is there any downsides or cases when it is not applicable?

**Main Review:**

# Originality

The work looks novel enough. The motivation is a bit lacking, though. A usual practical model is trained once and used for thousands of clients in production after. I would like to see a more extensive motivation why speeding up the training 1.2-1.5 times is important.

# Quality

In general, the work is focused on experimentation. The presented experiments look thorough. There are not enough details on how the training time was benchmarked. The work would benefit from computing number of FLOPS and comparing it the wall clock time. Second, all the experiments are focused on the language input. It would be beneficial to explore continuous tasks, such as time series.

It seems that the proposed approach depends on the mini-batch size. With a large mini-batch the speed up can be higher, but it might not be enough randomness in the mask sampling. With the small mini-batch I would expect small to no speed up. I am interested to see the discussion on this and experiments.

# Clarity

Section 3 is generally well written and easy to follow. The narrative is great. It is concise, but contains a lot of information. Section 4 is bit harder to follow, especially 4.3 looks rushed.

# Significance

The work looks significant enough.

# Suggestions for improvement

- Motivate the importance of this work.
- Compute FLOPS and describe the methodology of time benchmarking. Report the variance in time.
- Test with higher variety of data, such as time series, video, speech.
- Discuss and test the significance of the mini-batch size. Perhaps, use a toy dataset for this.

EDIT: the paper was improved, I increase the rating

**Time Spent Reviewing:**

5

---

> ### Author Response · Authors · 2021-08-10
> **Motivation & Importance; Mini batch size significance; Time benchmarking methodology; Non-NLP data; Limitations**
>
> **Motivation & Importance** :
>
> a. Due to sequential data dependency and complex interplay of gate structure, training and analyzing an LSTM network is inherently more complex than other DNN types. Our work facilitates step by step distillation of complex execution semantics and application of sparsity towards compute acceleration. This also serves as a template for analyzing sparsity opportunity for various other DNN types.
>
> b. Although training is considered only one time activity unlike inference; in reality training is also an repetitive process requiring several end to end runs in fine-tuning and finding optimal hyper-parameter settings for reaching target accuracy. In addition, each training run itself spans several hundreds of thousands of iterations which has non-negligible run-time and energy costs.
>
> c. For smaller dataset sizes, iso-accuracy Transformers tend to have significantly more parameters and required computation which makes LSTMs more attractive.
>
> d. Finally, emerging Graph Neural net class of applications[6] also use LSTMs as aggregation function, in addition to novel application deployment [7] which means efficient LSTM training still remains an important problem going forward.
>
> [6] "Inductive Representation Learning on Large Graphs" Hamilton & Ying et al. , NeurIPS 2017
>
> [7] “LSTM variants meet graph neural networks for road speed prediction” Lu et al. Neurocomputing 2020.
>
> **Mini batch size significance** :
>
> Thanks for raising this important point. Choice of mini-batch size is critical in DNN training and we note the following key points regarding the execution time and functional accuracy :
>
> 1. Increasing batch size increases latency per training step;
> 2. However, latency decreases per epoch due to reduced number of training steps per epoch
> 3. We also observe decreased statistical rate of convergence
>
> For Zaremba medium experiment, starting with baseline batch size of B=20, we perform experiments by doubling(2x) and quadrupling(4x) the batch size. We find that B=40 and B=80, latency per iteration increases by 8% and 77% respectively. However, epoch level latency reduces by 46% and 61% for the same two cases for the NR+Random. For structred drop case(NR+RH+ST), the per step increase in latency was found to be 20% and 60% respectively, while 60% and 85% reduction in per epcoh level latency.
>
> Regarding impact on accuracy, we find that acurracy worsenes for the same number of tranining epochs, which is a common trend for random as well as structured dropout application.
> More specifically, for NR+Random case, we observe that at 2x batch size, test perplexity(PPL - lower the better) increases by +4.7 and at 4x batch size, it further increases by +9.719.
> For the NR+RH+ST case, the increment in PPL at 2x and 4x batch sizes is +3.49 and +6.697, which is smaller than NR+Random case indicating that NR+RH+ST case is able to handle the accuracy degradation more gracefully.
>
> We will repeat these set experiments for the remaining workloads and configuration and conclude on the observation in our final version under appendix section.
>
> **Benchmarking methodology and variance in execution**:
>
> We follow a similar approach as by Wen et al [8] towards time benchmarking, which is obtained by measuring execution time around the most compute intensive portions of the workload, i.e., GEMM ops during FP, BP and WG stages. Since we perform experiments under controlled environments, there is a negligible variance(<2%) in running time across different runs.
>
> [8] : “Learning Structured Sparsity in Deep Neural Networks” Wen et al. NeurIPS 2016
>
> **Non-NLP datatypes** :
>
> Thanks for the suggestion. We performed dropout experiments with Time-Series data for Stock-Price prediction problem using 2 layer LSTM model with hidden sizes of 16/32. We observed that these LSTM networks’ hidden size tend to be smaller (usually <32) and in general the application of dropout isn't as effective as in the larger network scenario.  Also, while we  couldn't perform experiments with Speech and Video data due to time constraints,  we strongly believe our technique to be generally applicable towards large LSTM networks.   We are planning to include speech/video results in the final paper.
>
> **Limitation Discussion**:
>
> The proposed opportunity for speedup is limited by the scope of dropout itself - if dropout rate is small( e.g. ~0.1-0.2) or it is not applicable(e.g. small hidden size networks as discussed above), then speedup will not be appreciable/applicable.

---

### Official Review · Reviewer_nTft · 2021-07-16

**Rating:** 7
**Confidence:** 4

**Summary:**

This work is mainly concerned with the time cost of LSTM computation. Inspired by the sparsity of neural networks, the authors propose to structure dropout patterns by discarding the same collections of neural units within a training batch. They have also carefully studied how dropout-induced sparsity propagated through different training phrases and how it was utilized in each of them. The experiments are very solid, with the investigations on LM, NMT, and NER.

**Limitations And Societal Impact:**

1, Excellent paper, regarding its topic, idea, method, and experiments.

2, A discussion about Transformer is necessary. I may change the score if no convincing response is given.

**Main Review:**

The whole paper is really good. Accelerating the computation of sequential models is indeed important and the proposed method is awesome. However, the paper lacks a discussion on Transformer. For example, is your method applicable to improve its run-time efficiency? Transformer models are becoming more and more important in NLP. The authors should consider this.

**Time Spent Reviewing:**

5

---

> ### Author Response · Authors · 2021-08-10
> **Discussion on Transformers and Dropout applicability**
>
> Although transformers have recently gained traction in several NLP tasks, we believe that transformers serve useful for relatively large dataset problems, unlike LSTMs which can be readily adapted for small-medium sized datasets. Further, as also noted for the previous reviewer, starting with original transformer work [1], which employs dropout at residual output and at positional encoding for both encoder and decoder stack, our technique can be used as direct replacement. Prior works have further shown dropout applicability for transformers e.g. LayerDrop[2] which drops entire transformer layers vs DropHead[2] and DropMask[4], which propose to drop some of the attention heads within a transformer layer. While DropHead drops attention heads which are chosen at random, DropMask also explores schemes for dropping most important attention heads(instead of random) and a schedule for dropout rate during training. Note that dropping attention heads can be considered as an extension to our proposed column drop approach, with an added constraint of dropping consecutive columns corresponding to an attention head size. Thus, the proposed acceleration scope for FP, BP and WG stages remains equally valid in transformer context too, which is not considered by these prior works. And our proposed solution is a finer grain dropout mechanism that is likely to provide better speedup.
>
> [1] “Attention is all you need” Vaswani et al.,  NeurIPS 2017. (https://papers.nips.cc/paper/2017/file/3f5ee243547dee91fbd053c1c4a845aa-Paper.pdf)
>
> [2] “Reducing Transformer Depth on demand with Structured Dropout” Fan et al. (https://arxiv.org/pdf/1909.11556.pdf)
>
> [3] “Scheduled DropHead: A Regularization Method for Transformer Models” Zhou et al. (https://arxiv.org/pdf/2004.13342.pdf)
>
> [4] “Alleviating the Inequality of Attention Heads for Neural Machine Translation” Sun et al (https://arxiv.org/pdf/2009.09672.pdf)

---

### Official Review · Reviewer_S8Mq · 2021-07-16

**Rating:** 6
**Confidence:** 3

**Summary:**

This paper proposes a way to use dropout induced sparsity for LSTMs to reduce run-time in general purpose SIMD hardware and systolic arrays, by structuring dropout patterns. Experiments on LM (PTB dataset), MT (IWSLT De-En and En-Vi), and NER (CoNLL-2003) shows that the proposed approach can induce 1.23x to 1.64x speedup without sacrificing accuracy.


**Limitations And Societal Impact:**

Yes

**Main Review:**

* Pros
  * The core analysis of how to use dropout for efficient LSTM training is quite comprehensive
  * The general research direction to improve efficiency of NLP models is important
* Cons
  * This method seems only limit to RNN/LSTM models, yet RNN/LSTM is almost outdated. Can this method extends to Transformer?
  * The speedup of the proposed method is not significant (around 1.5x). How does it compare with constant optimization (i.e. better implementation)?



**Time Spent Reviewing:**

1

---

> ### Author Response · Authors · 2021-08-10
> **Dropout Applicability; Scope of improvement as compared to better implementation technique**
>
> **1. Applicability for transformers **
>
> The proposed methodology is not strictly limited to RNN/LSTM models and can be extensible to Transformer based models too. In fact, starting with original transformer work [1] , which employs dropout at residual output and at positional encoding for both encoder and decoder stack, our technique can be used as direct replacement. Further, there have been proposals for applying dropout to transformer models with different scopes e.g. LayerDrop[2], which drops entire transformer layers vs DropHead[3] and DropMask[4], which propose to drop some of the attention heads within a transformer layer. While DropHead drops attention heads which are chosen at random,  DropMask also explores schemes for dropping most important attention heads(instead of random) and a schedule for dropout rate during training. Note that dropping attention heads can be considered as an extension to our proposed column drop approach, with an added constraint of dropping consecutive columns corresponding to an attention head size. Thus, our proposed acceleration scope for FP, BP and WG stages remains equally valid in the transformer context too, which is not considered by the prior works that only focus on the regularization aspect of transformer learning.
>
> [1] “Attention is all you need” Vaswani et al.,  NeuIPS 2017 (https://papers.nips.cc/paper/2017/file/3f5ee243547dee91fbd053c1c4a845aa-Paper.pdf)
>
> [2] “Reducing Transformer Depth on demand with Structured Dropout” Fan et al. (https://arxiv.org/pdf/1909.11556.pdf)
>
> [3] “Scheduled DropHead: A Regularization Method for Transformer Models” Zhou et al. (https://arxiv.org/pdf/2004.13342.pdf)
>
> [4] “Alleviating the Inequality of Attention Heads for Neural Machine Translation” Sun et al (https://arxiv.org/pdf/2009.09672.pdf)
>
> **2.The speedup of the proposed method is not significant (around 1.5x). How does it compare with constant optimization (i.e. better implementation) **
>
> Previously proposed constant optimization technique for LSTM training, e.g. ECHO [5] obtains a speed-up factor of 1.28x working on EN-Vi dataset on the same Encoder-Decoder model discussed in Section 4.2 of our work. Therefore, speed-up improvements are comparable and within the scope of what is achieved by employing static optimization(e.g. better memory management) techniques. Such approaches are also orthogonal in nature indicating that overall improvements can be combined when jointly deployed. Also, our optimizations are obtained using CUDA GPU kernels, which is state of the acceleration library for NVIDIA GPU platforms.
>
> [5] “Echo: Compiler-based GPU Memory Footprint Reduction for LSTM RNN Training” Zheng et al, ISCA 2020 (https://ieeexplore.ieee.org/abstract/document/9138914)

---

> > ### Comment · Reviewer_S8Mq · 2021-08-26
> > **Response**
> >
> > Thanks for your response!
> >
> > For practical impact, I still think it will be beneficial to have results (not justifications) on Transformers, and ideally the speedup is greater than 1.23x to 1.64x (so not easily overshadowed by e.g. different implementations). But I like that it has thorough analysis and extensive experiments, which is probably more important for this venue. The rebuttal addressed my concerns to some extent(possibility to apply to Transformer, the speedup is not large but it's orthogonal to most other methods). So I will change my score to 6.

---

### Author Response · Authors · 2021-08-10
**Acknowledgment**

We would like to thank all reviewers for their insightful comments and suggestions. We are glad that the reviewers found the targeted research problem is important, the analysis and method is novel and comprehensive, the experiments are solid and thorough, and the paper being written well. We address all the concerns below including

1. Motivation and Applicability for Transformers
2. Impact of batch size
3. Methodology of time benchmarking
3. Discussion on other types of data in addition to language tasks
5. Test on different types of GPUs

We will incorporate all the comments in our final version.

---

### Decision · Program_Chairs · 2021-09-27

**Decision:**

Accept (Poster)

**Comment:**

Paper presents results on structured dropout in recurrent models that can be used to speed up computation without affecting results. Reviewers found sufficient depth of exploration in results, and the results looked decent. Reviewers wanted to hear about the applicability to transformers, and the authors addressed that in the rebuttal.